# A Multiple Mechanism Enhanced Arithmetic Optimization Algorithm for Numerical Problems

**DOI:** 10.3390/biomimetics8040348

**Published:** 2023-08-06

**Authors:** Sen Yang, Linbo Zhang, Xuesen Yang, Jiayun Sun, Wenhao Dong

**Affiliations:** College of Information and Communication Engineering, Harbin Engineering University, Harbin 150001, China; zhanglinbo@hrbeu.edu.cn (L.Z.); yxsky@hrbeu.edu.cn (X.Y.); jiayunsun@hrbeu.edu.cn (J.S.); dongwenhao@hrbeu.edu.cn (W.D.)

**Keywords:** arithmetic optimization algorithm, meta-heuristic algorithm, global optimization, exploration and exploitation, wireless sensor coverage

## Abstract

The Arithmetic Optimization Algorithm (AOA) is a meta-heuristic algorithm inspired by mathematical operators, which may stagnate in the face of complex optimization issues. Therefore, the convergence and accuracy are reduced. In this paper, an AOA variant called ASFAOA is proposed by integrating a double-opposite learning mechanism, an adaptive spiral search strategy, an offset distribution estimation strategy, and a modified cosine acceleration function formula into the original AOA, aiming to improve the local exploitation and global exploration capability of the original AOA. In the proposed ASFAOA, a dual-opposite learning strategy is utilized to enhance population diversity by searching the problem space a lot better. The spiral search strategy of the tuna swarm optimization is introduced into the addition and subtraction strategy of AOA to enhance the AOA’s ability to jump out of the local optimum. An offset distribution estimation strategy is employed to effectively utilize the dominant population information for guiding the correct individual evolution. In addition, an adaptive cosine acceleration function is proposed to perform a better balance between the exploitation and exploration capabilities of the AOA. To demonstrate the superiority of the proposed ASFAOA, two experiments are conducted using existing state-of-the-art algorithms. First, The CEC 2017 benchmark function was applied with the aim of evaluating the performance of ASFAOA on the test function through mean analysis, convergence analysis, stability analysis, Wilcoxon signed rank test, and Friedman’s test. The proposed ASFAOA is then utilized to solve the wireless sensor coverage problem and its performance is illustrated by two sets of coverage problems with different dimensions. The results and discussion show that ASFAOA outperforms the original AOA and other comparison algorithms. Therefore, ASFAOA is considered as a useful technique for practical optimization problems.

## 1. Introduction

With the rapid development in various fields, real-world optimization problems are becoming more and more complicated. When dealing with these emerging optimization problems, traditional optimization methods require too much time and expensive. In most cases, it is known that relatively exact solutions are acceptable, meaning that the estimated better solution is acceptable in production practice. Metaheuristic algorithms are an emerging class of optimization techniques with the advantages of high operational efficiency, flexibility, stability, simplicity of implementation, parallelism, and ease of combining with other algorithms [1]. Therefore, many optimization algorithms have been proposed in recent decades to solve these nonconvex, nonlinearly constrained, and complex optimization problems and have proven to be very effective for these practical problems.

As one of the novel algorithms, AOA was initially applied to numerical optimization problems and engineering design problems. Due to its uncomplicated structure and excellent performance, AOA has covered many areas such as support vector regression (SVR) parameter optimization [2], tuning PID controllers [3,4], fuel cell parameter extraction [5], DNA sequence optimization design [6], clustering optimization [7,8], power system stabilizer design [9], feature selection [10], photovoltaic parameter optimization [11,12,13], robot path planning [14], wireless sensor network location and deployment [15], IoT workflow scheduling [16], image segmentation [17], etc.

Although AOA has better performance, it reduces the convergence rate and tends to fall into local optimum solutions when facing optimization problems with complex structures. Therefore, the convergence accuracy and convergence speed of AOA are achieved by adopting various mechanisms. For example, Dhawale et al. used the Levy flight strategy to enhance the exploitation and exploration capabilities of AOA [18]. Zhang et al. proposed a hybrid AOA algorithm that introduces energy parameters for Harris hawks optimization to balance exploitation and exploration [19]. Izci et al. proposed a hybrid arithmetic optimization algorithm incorporating a Nelder–Mead simplex search for the optimal design of automotive cruise control systems [20]. Chen et al. proposed an improved algorithmic optimization algorithm based on a population control strategy that classifies populations and adaptively controls the number of individuals in subpopulations, effectively using information about each individual to improve the accuracy of the solution [21]. Davut et al. modified the basic opposites learning mechanism and applied it to enhance the population diversity of arithmetic optimization algorithms [22]. Fang et al. used dynamic inertia weights to improve the exploration exploitation capability of the algorithm and introduced dynamic variance probability coefficients and triangular variance strategies to help the algorithm avoid local optima. Zhang et al. used a differential variance ranking strategy to improve the local exploitation capability of AOA [23]. Abualigah et al. mixed AOA with the sine and cosine algorithm to enhance the local search performance of the algorithm [24]. Celik et al. introduced Gaussian distribution and quasi-opposite learning strategy in order to improve the deficiency of slow convergence of AOA [25]. Ozmen et al. presented an augmented arithmetic optimization algorithm integrating pattern search and elite opponent learning mechanisms [26]. Zheng et al. instead used stochastic mathematical optimizer probabilities to increase population diversity and proposed a forced switching mechanism to help populations jump out of local optimum [27]. An improved arithmetic optimization algorithm combining the logarithmic spiral mechanism and the greedy selection mechanism was proposed and employed for solving PID control problems by Ekinci et al. [28].

This work presented a variant of AOA called ASFAOA for numerical optimization and wireless sensor coverage problems by integrating a double-opposite learning mechanism, an adaptive spiral search strategy, an offset distribution estimation strategy, and a modified cosine acceleration function formulation into the original AOA. The contributions of this work are summarized as follows:-The double-opposed learning strategy was used to enhance population diversity. As a result, the global exploration capability of the method was improved.-The adaptive spiral search strategy was used to adequately search the space around each individual, and thus the local optimal avoidance of the method was further improved.-The offset distribution estimation strategy is used to efficiently utilize the dominant population information to guide the individuals towards correct evolution. Thus, the accuracy of the method is further improved.-The adaptive cosine acceleration function is used to balance the exploitation and exploration ability of the algorithm. Thus, the convergence speed and accuracy of the populations are accelerated.-ASFAOA was evaluated on the CEC2017 benchmark function to validate its global optimization capability.-ASFAOA is used to solve the wireless sensor coverage problem.

The structure of the article is set as follows: Section 2 gives an overview of the original AOA. Section 3 describes the implementation of the improved method, among other specifics. The results and discussion of comparing ASFAOA with other algorithms on CEC2017 function test and wireless sensor coverage problems are shown in Section 4. Finally, Section 5 shows the conclusions of the proposed work and future plans.

## 2. Arithmetic Optimization Algorithm

The AOA is a new, meta-heuristic method proposed by Abualigah in 2021 [29]. The AOA utilizes four traditional arithmetic operators to build position update formulas. The specific formulas are presented separately as follows:

### 2.1. Initialization Phase

In AOA, the initial population is generated randomly in the search space with the following equation:(1)$$X_{i}^{\mathrm{int}}=rand\times (ub-lb)+lb,i=1,2,...,NP$$
where $X_{i}^{\mathrm{int}}$ is the $i^{th}$ initial individual, ub and lb are the upper and lower boundaries of the search space. NP is the number of tuna populations. rand is a uniformly distributed random vector ranging from 0 to 1.

After initializing the population, Math Optimizer Accelerated (MOA) is computed to choose whether to perform exploitation or exploration behavior.
(2)$$MOA=min+t\times (\frac{max-min}{t_{max}})$$
where t and $t_{\max}$ denote the current iteration and the maximum iteration. max and min are 0.9 and 0.2.

### 2.2. Exploration Phase

When MOA > 0.5, the problem is explored globally using multiplication and division operators. The mathematical model is as follows.
(3)$$X_{i}^{t+1}=\{\begin{array}{l} X_{b}^{t}÷(MOP+eps)\times ((ub-lb)\times \mu +lb),rand<0.5\\ X_{b}^{t}\times MOP\times ((ub-lb)\times \mu +lb),rand\geq 0.5 \end{array}$$
where $X_{b}^{t}$ is the global best agent. eps is a minimal value that guarantees that the denominator is not zero. μ is a constant with value of 0.499.

The Math Optimizer probability (*MOP*) is as follows:(4)$$MOP=1-{\left(\frac{t}{t_{max}}\right)}^{0.2}$$

### 2.3. Exploitation Phase

When MOA < 0.5, the exploitation is performed using operators (subtraction (“−”) and addition (“+”)). The mathematical model is as follows:(5)$$X_{i}^{t+1}=\{\begin{array}{l} X_{b}^{t}-MOP\times ((ub-lb)\times \mu +lb),rand<0.5\\ X_{b}^{t}+MOP\times ((ub-lb)\times \mu +lb),rand\geq 0.5 \end{array}$$

## 3. The Proposed ASFAOA

The basic arithmetic optimization algorithm is a simple, structured algorithm with some power of searching for the best individual, but there are still several weaknesses as follows. First, in terms of population initialization, AOA randomly initializes populations in the search space and does not spread the entire search space well. For the four updated methods of AOA, all of them focus only on the best individual of the group and update the random position around the best individual according to the random value. Where there is a lack of information exchange between individuals, “eyes” only focus on “the current optimal position of the group” regardless of the rest of the search individuals, so that the individual search efficiency is extremely low. This will inevitably cause the search agents to over-gather in the vicinity of the current optimal position of the population, causing the diversity of the population to be ineffectively maintained and the algorithm to fall into a local optimum.

AOA selects exploitation or exploration by controlling the change in the acceleration function MOA. The larger the MOA, the greater the global search capability of the algorithm. the smaller the MOA, the greater the local exploitation capability of the algorithm. In the basic AOA algorithm, MOA grows linearly, which means the global search capability of the algorithm increases linearly. This is inconsistent with the common search strategy of swarm intelligence optimization algorithms, where the algorithm focuses on global exploration in the early stage of search and local exploitation in the later stage. On the other hand, the AOA is non-linear in evolutionary exploration, and the linear growth of MOA cannot accurately approximate the actual iterative process, which makes it difficult for the AOA algorithm to balance exploitation and exploration.

To improve the shortcomings of the basic arithmetic optimization algorithm and enhance its performance, this paper proposes an AOA variant called ASFAOA. The optimization performance of AOA is enhanced by the following approaches. First, the population diversity is enhanced by initializing the population using double-opposition learning, as well as enhancing the population diversity by more effectively searching the problem space using the double-opposed learning strategy. Second, the spiral search strategy of the tuna swarm optimization algorithm is introduced into the addition and subtraction strategy of AOA so as to search the space around each individual more effectively, thus enhancing the ability of AOA to jump out of the local optimum. In the third, an adaptive cosine acceleration function is proposed to better balance the exploitation and exploration capabilities of the algorithm. Fourth, an offset distribution estimation strategy is used to effectively utilize the dominant population information to guide individuals to evolve correctly. In the fifth, a stochastic boundary control strategy is proposed to increase the search range of each individual. The details of the improvement strategy are described as follows.

### 3.1. Double-Opposition Learning Strategy (DOL)

The opposition learning strategy is a new technique that has emerged in the field of optimal computing in recent years. The opposition learning strategy mainly enhances population diversity and avoids the algorithm from falling into local optimum by generating the inverse position of each individual and evaluating the original and opposition individuals to retain the dominant individual into the next generation. The specific formula is as follows:(6)$$X_{i}^{o}=lb+ub-X_{i}^{t}$$
where $X_{i}^{o}$ is the corresponding opposite solution of $X_{i}^{t}$. In order to further enhance the population diversity and overcome the deficiency that the opposed solution generated by the basic opposed learning strategy is not necessarily better than the current solution, considering that the tent chaos mapping has the characteristics of randomness and ergodicity, which can help generate new solutions and enhance the population diversity [30], this paper combines the tent chaos mapping with the opposed learning strategy and proposes a tent opposite-learning mechanism. The specific mathematical model is described as follows:(7)$$X_{i}^{To}=lb+ub-\lambda_{i}\cdot X_{i}^{t}$$
(8)$$\lambda_{i+1}=\{\begin{array}{cc} 2\lambda_{i}, & 0\leq \lambda_{i}\leq 0.5\\ 2(1-\lambda_{i}), & 0.5<\lambda_{i}\leq 1 \end{array}$$
where, $X_{i}^{To}$ denotes the solution generated by the tent opposition learning corresponding to the i individual in the population. $\lambda_{i}$ is the corresponding tent chaotic mapping value. In addition to the tent opposition learning strategy, a lenticular opposition learning strategy is also proposed. This strategy uses the property that an individual will become an inverted real image on the other side of the convex lens when it is out of focus to build a mathematical model, as follows:(9)$$X_{i}=\frac{lb+ub}{2}+\frac{lb+ub}{2k}-\frac{X_{i}}{k}$$
where *k* is the scaling factor, and the value of *k* affects the quality of the generation of the opposite solution. The smaller the value of *k*, the larger the range of the generated opposite solution. The larger the value of *k*, the smaller the range of the opposite solution that can be provided. Considering that the algorithm performs a more global search in the early stage and more exact exploitation in the later stage, a dynamically adjustable scaling factor formula is proposed as shown below:(10)$$k={\left(1+{\left(\frac{t}{t_{\max}}\right)}^{1/3}\right)}^{c}$$
where c is a constant with a value of 3.

### 3.2. Adaptive Spiral Search Strategy (ASS)

In the local exploitation phase of AOA, AOA performs random position updates around the optimal individual, which is beneficial to the fast convergence of the algorithm, but when the optimal solution falls into a local optimum, it easily leads to other individuals following into the local optimum. To protect the AOA algorithm from falling into a local optimum, the spiral foraging strategy, inspired by the spiral foraging strategy of the tuna swarm algorithm [31], is introduced into the addition and subtraction operations in AOA. The AOA randomly selects one of the strategies from the original strategy and the spiral foraging strategy to update the individual positions. The spiral foraging strategy is specified as follows:(11)$$X_{i}^{t+1}=\{\begin{array}{l} \alpha_{1}\cdot (X_{br}^{t}+\beta \cdot \left|X_{br}^{t}-X_{i}^{t}\right|)+(1-\alpha_{1})\cdot X_{i}^{t},\;i=1\\ \alpha_{1}\cdot (X_{br}^{t}+\beta \cdot \left|X_{br}^{t}-X_{i}^{t}\right|)+(1-\alpha_{1})\cdot X_{i-1}^{t}\;,i=2,3,...,NP \end{array}$$
(12)$$\alpha_{1}=a+(1-a)\cdot \frac{t}{t_{\max}}$$
(13)$$\beta =e^{bl}\cdot \cos(2\pi b)$$
(14)$$l=e^{3\cos((\frac{t_{\max}+1}{t}-1)\pi )}$$
where a is a constant value of 0.7 and b is a random number uniformly distributed from 0 to 1. $X_{br}^{t}$ denotes the optimal individual or a randomly generated individual in the search space. In the global exploration phase of AOA, the spiral search strategy needs to search a wider space, so $X_{br}^{t}$ is a randomly generated individual in the search space. In the later local development phase, the spiral search is more focused around the optimal individual, so $X_{br}^{t}$ takes the location information of the optimal individual. Each individual is chosen to use either the original search strategy or the spiral search strategy at each iteration, according to the probability pr=rand.

### 3.3. Adaptive Cosine Acceleration Function (ACA)

When AOA uses MOA to switch between “global exploration” and “local exploitation”, the probability of local exploitation in the later stage of the search is lower than that of global exploration, which weakens the ability of local exploitation in the later stage of the algorithm and is not conducive to the optimization of the algorithm. On the other hand, the AOA is nonlinear in the evolutionary exploration process, and the linear growth of MOA cannot accurately approximate the actual iterative process, so the introduction of cosine control factor converts the change of MOA to nonlinear, which can more closely match the actual iterative process of the algorithm.
(15)$$MOA=\min+(\max-\min)\times \cos^{2}(\frac{\pi t}{2t_{\max}})$$

Figure 1 shows the comparison between the original *MOA* and the improved *MOA*. From the figure, we can see that the improved *MOA* in this paper maintains a large value at the beginning of the algorithm iteration, which enables the algorithm to perform global search adequately; at the later part of the iteration, the *MOA* rapidly decreases to a smaller value, which increases the local exploitation probability of the algorithm and improves the convergence speed of the algorithm.

### 3.4. Offset Distribution Estimation Strategy (ODE)

Analyzing the basic AOA algorithm, we can see that each individual mainly follows the optimal individual for position updating. When the optimal individual falls into a local optimum, it will cause the rest of the individuals to fall into the same local optimum. At the same time, there is a lack of mutual information exchange between each individual. In order to enhance the population diversity, strengthen the information exchange among individuals, and improve the algorithm’s performance in finding the optimal, this paper introduces the offset distribution estimation strategy. The distribution estimation strategy represents the relationship between individuals through a probabilistic model [32,33]. Assuming that the problem model obeys a multivariate Gaussian probability distribution, the distribution model is computed using half of the individuals of the current population and sampling new offspring to drive the optimization process of the algorithm. The basic computational process can be divided into the following four steps:(1)Set the algorithm parameters and initialize the population;(2)Evaluate the solutions according to the objective function values;(3)Select the partially optimal solutions to compute the Gaussian probability distribution model;(4)Sample the new population according to the updated probability model; repeat step 2 until the end condition is satisfied.

The strategy uses the current dominant population to calculate the probability distribution model and generates new child populations based on the sampling of the probability distribution model, and finally obtains the optimal solution through continuous iteration. In this chapter, half of the better performing populations were selected for sampling, and the mathematical model of this strategy is described as follows.
(16)$$newX_{i}^{t+1}=m+randn\cdot (m-X_{i}^{t})$$
(17)$$m=(X_{b}^{t}+X_{mean}^{t}+X_{i}^{t})/3$$
(18)$$Cov=\frac{2}{N}\sum_{i=1}^{Np/2}(X_{i}^{t+1}-X_{mean}^{t})\times {\left(X_{i}^{t}-X_{mean}^{t}\right)}^{T}$$
(19)$$X_{mean}^{t}=\sum_{i=1}^{Np/2}\omega_{i}\times X_{i}^{t}$$
(20)$$\omega_{i}=\frac{\ln(Np/2+0.5)-\ln(i)}{\sum_{i=1}^{N/2}(\ln(Np/2+0.5)-\ln(i))}$$
where $X_{mean}^{t}$ is the weighted covariance matrix of the dominant population, $\omega_{i}$ is the weighting coefficient of the dominant population in descending order of fitness value, and Cov is the weighted covariance matrix of the dominant population. By considering the optimal individual information, the dominant population weighted information and its own information, the evolutionary direction of the population is corrected to improve the performance of the algorithm in the search for superiority.

### 3.5. Boundary Control Strategy

When the agent position is beyond the search space, it is usually to reinitialize the individual at the boundary, but this tends to derive multiple agents at the boundary position, which is not conducive to the exploration of the population in the whole search space. In order to increase the search range of each agent, this paper proposes a randomized boundary control strategy, which randomly generates the dimensional information in the whole search space when a dimension of the agent is beyond the search boundary, and the specific mathematical model is shown as follows:(21)$$X_{i,j}^{t}=\{\begin{array}{ll} lb+rand\cdot (ub-lb), & if\;X_{i,j}^{t}>ub\;or\;X_{i,j}^{t}<lb\\ X_{i,j}^{t}, & if\;X_{i,j}^{t}\leq ub\;and\;X_{i,j}^{t}\geq lb \end{array}$$
where j denotes the *j*-th dimension of each individual.

### 3.6. Pseudo-Code of ASFAOA

The pseudo code of the ASFAOA is shown in Algorithm 1.
**Algorithm 1** Pseudo-code of the ASFAOA algorithm.1Initialize the Arithmetic Optimization Algorithm parameters α, µ2Initialize the parameters *a*3Initialize the solutions’ positions randomly. (Solutions: *i* = 1,...,Np)4**while** (t < t_max_) **do**5 Calculate the Fitness Function for the given solutions6 Find the best solution (Determined best so far).7 Update the MOA value using Equation (15).8 Update the MOP value using Equation (4).9 Update the *k* value using Equation (10).10 Update the *Cov* value using Equations (18)~(20).11 **for** (*i* = 1 to Solutions) **do**12  Update positions by Equations (7) and (9)13  Generate a random values between [0, 1] (r1, r2, and r3)14  **if** r1 > MOA **then**15   **if** r2 > 0.5 **then**16     Update positions by Equation (3)17   **else**18     Update positions by Equation (16)19   **end if**20  **else**21   **if** r3 > 0.5 **then**22     Update positions by Equation (5)23   **else**24     Update positions by Equation (11)25   **end if**26  **end if**27  Update positions by Equation (21)28 **end for**29 *t* = *t* + 130**end while**31Return the best solution.

### 3.7. The Computational Complexity of ASFAOA

The time complexity of AOA can be seen from the literature [25] as follows.
(22)O(AOA)=O(T(O(Exploration Phase+Exploitation Phase)))
(23)O(AOA)=O(T(Np·D+Np·D))=O(T·Np·D)

In this paper, five improvement strategies are proposed; *ASS* and *ACA* do not change the time complexity. The time complexity of the covariance matrix of *ODE* is $O(T(Np/2\cdot D^{2}))$, and the *DOL* is O(T·Np·D). Therefore, the time complexity of m-*ASFAOA* is shown below.
(24)O(ASFAOA)=O(T(O(Exploration Phase+ODE)+O (Exploitation Phase+ASS))+O(DOL)))
(25)$$\begin{array}{cc} O(ASFAOA) & =O(T(Np/2\cdot D+Np/2\cdot D)+T(Np/2\cdot D+Np/2\cdot D^{2})+T(Np\cdot D))\\ & =O(T\cdot NP/2\cdot D^{2}+T\cdot Np\cdot D) \end{array}$$

## 4. Experimental Results and Discussion

To demonstrate the superiority of the proposed ASFAOA, two different experiments are conducted in this section, including CEC2017 benchmark function test and wireless sensor coverage optimization. During the experiments, the proposed approach is also compared with the current well-known techniques. All experimental results and discussions validate the competitive performance of the proposed ASFAOA in solving various optimization problems.

### 4.1. CEC2017 Benchmark Functions Test

In this section, the performance of the proposed ASFAOA is evaluated with the CEC2017 test function. Many meta-heuristic algorithms design update formulas for classical test functions, which then achieve better optimization performance. CEC2017 has a more complex structure and more difficult to solve compared to these functions, which allows for a better validation of the algorithm’s performance. The details of the test function are presented first. Then, six metaheuristics are used to illustrate the outstanding performance of the proposed improved algorithm. The convergence accuracy of the algorithms is analyzed in two aspects, including the mean and standard deviation. The mean value is the average solution obtained from these tests. The standard deviation is used to reflect the dispersion of the optimal solution. In addition, statistical methods such as Wilcoxon signed rank test and Friedman ranking test were used to confirm significant differences between ASFAOA and other algorithms. Convergence curves and box plots are used for a visual description of the optimization effect.

#### 4.1.1. Experimental Settings

The IEEE CEC2017 test function includes 28 benchmark functions whose feasible range is [−100, 100]. The specific content of CEC 2017 is shown in Table 1. The Fi * denotes theoretical optimum of each function.

In this test, six typical swarm intelligence algorithms are selected to take part in the comparison test, such as the whale optimization algorithm (WOA) [34], sine cosine algorithm (SCA) [35], Harris hawks optimization (HHO) [36], sparrow search algorithm (SSA) [37], tunicate swarm algorithm (TSA) [38], butterfly optimization algorithm (BOA) [39]. The parameters of the mentioned algorithms are displayed on Table 2.

To perform a proper comparison, the number of iterations and population size were set to 500 and 600, respectively. Each algorithm was tested 51 times independently to obtain reliable statistical results.

#### 4.1.2. 30D Functions Test Results and Analysis

In the study, ASFAOA was compared with six swarm intelligence algorithms on the CEC2017 function with D = 30. The specific results of the experiment are shown in Table 3. These data were calculated from the optimal values obtained by solving different functions 51 times. On each function, all algorithms are sorted, and the smaller the optimal value, the smaller the ranking. As can be seen, it mainly includes the mean value, standard deviation and ranking based on the two mentioned indicators. In the meantime, the last two rows in Table 3 indicate the average and final ranking of all algorithms. The proposed ASFAOA obtained the best ranking value of 1.04, which is the top one. HHO obtained the next-best ranking value of 2.71. SSA had the worst ranking of 6.86. It is worth noting that AOA scored 5.61, which is much larger than ASFAOA. Specifically, ASFAOA outperformed AOA on all functions and outperformed the rest of the comparison algorithms in 27 out of 28.

Table 4 illustrates the *p*-values calculated by the Wilcoxon singed-rank test for each function of each algorithm. If the value is less than 0.05, it means that there is a significant difference between ASFAOA and the other competitors, otherwise there is no significant difference. It can be seen that ASFAOA is significantly different from the other algorithms for most functions. The last row of Table 4 shows the results of comparing ASFAOA with other methods. ASFAOA performs worse than BOA on F19 and better than other methods on 27 functions. It is noteworthy that ASFAOA outperforms it on all functions compared to AOA. Furthermore, the average ranking results obtained with the various methods according to the Friedman test are shown in Figure 2. As shown in Figure 2, ASFAOA obtained the best average ranking value of 1.03, HHO ranked second with a value of 2.71, WOA and SCA followed HHO, and AOA ranked sixth. The results of the Friedman test further prove that ASFAOA outperforms other algorithms with significant advantages.

The convergence curves of all algorithms with different functions are shown in Figure 3. It can be seen that the enhanced ASFAOA performs the best compared to other methods. Specifically, ASFAOA has the fastest convergence speed and better convergence accuracy on F1–F5, F7, F9–F12, F15–F18, and F20–F28. In solving the rest of the functions, ASFAOA can achieve higher convergence accuracy in the later phase although the convergence speed is slower in the early phase. Generally, the convergence performance of ASFAOA is better than the comparison algorithm, especially compared to AOA. This phenomenon can be attributed to the following factors: on the one hand, the offset distribution estimation strategy guides the search agents to evolve towards more promising regions at a faster rate. On the other hand, the dual-opposed learning strategy and the spiral search strategy help to further improve the accuracy of the solution as well as the population diversity. In addition, the modification of MOA makes the algorithm more balanced in terms of exploitation and exploration capabilities.

To analyze the distribution characteristics of the solutions solved by ASFAOA, box plots were drawn based on the results of 51 independent solutions for each algorithm, as shown in Figure 4. For each algorithm, the center mark of each box indicates the median of the results of 51 times solved function. The bottom and top edges of the box indicate the first and third quartile points. The symbol “+” indicates bad values that are not inside the box. We can learn from Figure 3 that ASFAOA has no outliers when solving nine of the test functions (F1, F2, F8, F11, F14, F16, F24–F26), which indicates that the distribution solved with ASFAOA is very concentrated. For other test functions with bad values (F2, F4–F5, F7–F8, F12–F13, F15–F18, F20–F21, F24, F26), ASFAOA has a smaller median, which indicates that the quality of the solutions of ASFAOA is relatively better. Therefore, the improved algorithm proposed in this paper is robust.

#### 4.1.3. Analysis of ASFAOA Improvement Strategies

In this paper, the proposed improvement method for basic AOA consists of four parts: offset distribution estimation strategy (ODE), adaptive cosine acceleration function (ACA), adaptive spiral search strategy (ASS), and double-opposition learning strategy (DOL). To evaluate the effectiveness of different modification strategies, we present four variants of ASFAOA using different modification strategies as shown in Table 5. ASFAOA-1 utilizes DOL strategy to improve the algorithmic performance. ASFAOA-2 serves for evaluating the effectiveness of the ASS strategy. ASFAOA-3 utilizes the ACA strategy to balance algorithm exploitation and exploration capabilities. ASFAOA-4 incorporates the ODE strategy. The performance of the six algorithms was compared using the CEC2017 test suite. Each function was run independently 51 times. Table 6 lists the average error results for each algorithm, and the last row gives the Friedman test results for the six algorithms.

Significantly, ASFAOA with a complete improvement strategy performed the best, with a ranking of 1.04 in the Friedman test. The four derived algorithms, having one of the methods, respectively, also ranked better than the basic AOA. The four derived algorithms are ranked as 4.04, 3.29, 3.00, and 4.07. Hence, it can be concluded that the impact of these four modifications on the performance in descending order is: ODE > ASS > DOL > ACA. ASFAOA-3 performs the best among the four derived algorithms, proving that the utilization of ODE can effectively improve the performance. It generates offspring by utilizing the overall distribution information of the dominant population, which effectively avoids the defect that the population only follows the optimal individuals and falls into local optimum. ASFAOA-2 performs similarly to ASFAOA-3 and ranks third. This is due to the adoption of the ASS strategy, which randomly selects an individual as a reference point in the early stage, effectively broadening the search range and enhancing the algorithm’s ability to solve multimodal functions. In the later period, the optimal individual is selected as the reference point, and the search range is narrowed by adaptive narrowing to ensure the convergence efficiency. ASFAOA-1 strengthens population diversity by generating reverse individuals, and the experimental results also illustrate that this strategy is effective. ASFAOA-4 achieves improved performance by simply modifying the control parameters of the original algorithm, suggesting that the method strikes a certain degree of balance between exploitation and exploration behaviors.

### 4.2. Wireless Sensor Coverage Optimization Test

In this section, the performance of the proposed ASFAOA is evaluated using the wireless sensor coverage optimization problem. The details of the wireless sensor coverage problem are first presented. Then, the superior performance of the proposed algorithm is illustrated using the better-performing comparison algorithm in Section 4.1. The convergence accuracy of the algorithm is analyzed in terms of the optimal value, the mean value, and the standard deviation.

#### 4.2.1. Mathematical Models

In the wireless sensor network, the set of homogeneous wireless sensor nodes is $S=\{s_{1},s_{2},s_{3},...,s_{i},,,,s_{N}\}$; the sensing radius is *R_s_* and the monitoring area is a rectangular area of L×W. For calculation purposes, the rectangular area is discretized into L×W grids of equal area. The monitoring point is located at the geometric center of the grid. If the distance between the monitoring point and any node is less than or equal to the sensing radius $R_{s}$, the monitoring point is considered to be covered by the wireless sensor network. The set of monitoring nodes is $M=\{m_{1},m_{2},m_{3},...,m_{j},...,M_{L\times W}\}$. $\left(x_{i},y_{i}\right)$ and $\left(x_{j},y_{j}\right)$ correspond to the two-dimensional spatial coordinates of $s_{i}$ and $m_{j}$ in the set, respectively. The Euclidean distance between the two nodes is as below:(26)$$d(s_{i},m_{j})=\sqrt{{\left(x_{i}-x_{j}\right)}^{2}+{\left(y_{i}-y_{j}\right)}^{2}}$$

The probability of monitoring point $m_{j}$ being sensed by node $s_{i}$ is defined as:(27)$$p_{cov}(s_{i},m_{j})=\{\begin{array}{l} 1,\;\;if\;d(s_{i},m_{j})\leq R_{s}\\ 0,\;\;otherwise \end{array}$$

The wireless sensor coverage problem can be solved using either a multi-objective optimization algorithm or a single-objective optimization algorithm depending on the factors considered [40,41]. In this paper, we focus on verifying the superiority of the proposed algorithms and hence mainly solve the problem with coverage as the objective. We define the area coverage $C_{r}$ of all sensor nodes in the target monitoring environment as the ratio of the area covered by the set of sensor nodes to the area of the monitoring area.
(28)$$C_{r}=\frac{\sum_{j=1}^{L\times W}p_{cov}(s_{i},m_{j})}{L\times W}$$

#### 4.2.2. Simulation and Analysis

Two sets of experiments were designed to verify the ASFAOA performance. To make the experimental data more convincing, the simulation experiments for each algorithm were conducted 30 times independently, and the optimal value, mean and standard deviation were taken as statistics for comparison. In each experiment set, all parameter settings are the same for all four methods. The algorithms involved in the simulations include ASFAOA, AOA, HHO, and WOA.

##### Case 1

In Case 1, the monitoring area is a 10 m × 10 m two-dimensional square plane. The number of sensor nodes is 25, the sensing radius is 1, and the communication radius is 2. Table 7 shows the statistics of the optimization results by each algorithm. Figure 5 shows the iterative convergence curve of coverage optimization, and Figure 6 shows the node deployment of WSN after optimization of each algorithm.

As can be seen from Table 5, ASFAOA achieves a 15.7%, 8.26%, and 8.26% improvement in coverage compared with the optimized results of AOA, HHO, and WOA, respectively. Additionally, ASFAOA has better stability. From Figure 5, it can be observed that ASFAOAs do not fall into the local optimum and achieve a better coverage solution at the end, although the convergence speed is slow in the early phase. From the optimized node deployment of each algorithm in Figure 6, the node deployment after AOA optimization has a larger coverage blind area, while the result of ASFAOA optimization makes the sensor nodes more uniformly distributed, which verifies the effectiveness of the improved strategy.

##### Case 2

In case 2, the experimental settings are a two-dimensional plane with a monitoring area of 50 m × 50 m. The number of sensor nodes is 35. The sensing radius is 2.5 and the communication radius is 5. Table 6 records the optimization results for case 2. The coverage convergence curve for each algorithm is shown in Figure 7. Deployment scheme of nodes for each algorithm is shown in Figure 8.

According to the analysis of Table 8, ASFAOA continues to maintain its high performance in the optimization of Case 2 and finally achieves an average coverage rate of 83.93%. Compared with the optimization results of AOA, HHO, and WOA, the coverage rate improved by 14.53%, 5.11%, and 5.11%, respectively. In addition, the minimum mean value of ASFAOA indicates its better performance. As can be seen from Figure 7, ASFAOA can effectively avoid falling into local optimum and achieve a better coverage solution. By Figure 8, it can be seen that the optimal coverage solution obtained by ASFAOA has a more uniform distribution of nodes. The solution finally given by ASFAOA improves the coverage rate from 65 to 83.93, which indicates that the algorithm proposed in this paper has excellent search capability.

## 5. Conclusions

In this paper, we have proposed an improved variant of AOA, named ASFAOA for the global optimization problem. The convergence accuracy and convergence speed of ASFAOA are supported by the dual-opposition learning strategy, the adaptive spiral search strategy, and the offset distribution estimation strategy. In order to validate and analyze the superiority of ASFAOA, a large number of experiments were conducted, including the CEC 2017 test suite of mean analysis, convergence analysis, stability analysis, statistical tests, and two sets of wireless sensor coverage problems with different dimensions. The results and discussions validate the rationality and usability of the improved strategies. The application of ASFAOA to wireless sensor coverage problems verifies the capability of ASFAOA in solving practical optimization problems.

In the future, we will focus our attention on two subsequent directions: One is to further investigate the internal mechanisms of ASFAOA with the aim of reducing its computational complexity and improving its performance. The other is to develop a multi-objective version of ASFAOA for solving more practical problems such as robot path planning, optimal control of electric vehicle composite braking, multilevel thresholding image segmentation, UAV mission planning, etc.

## Figures and Tables

**Figure 1 biomimetics-08-00348-f001:**
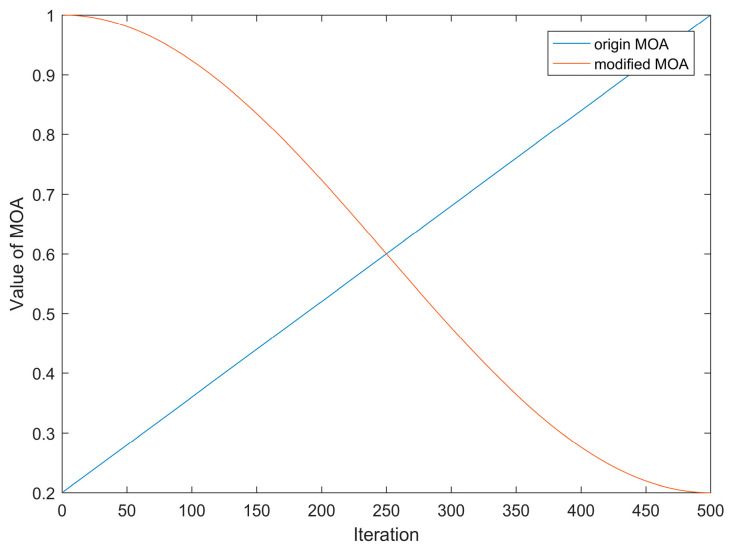
MOA dynamic variation curve.

**Figure 2 biomimetics-08-00348-f002:**
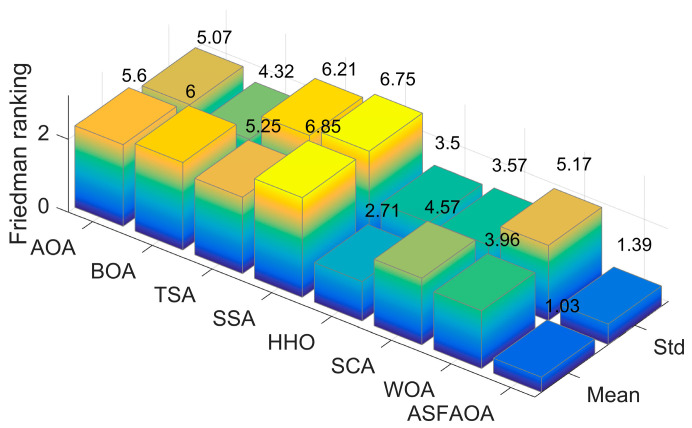
Friedman ranking of different approaches on CEC2017 30D test.

**Figure 3 biomimetics-08-00348-f003:**
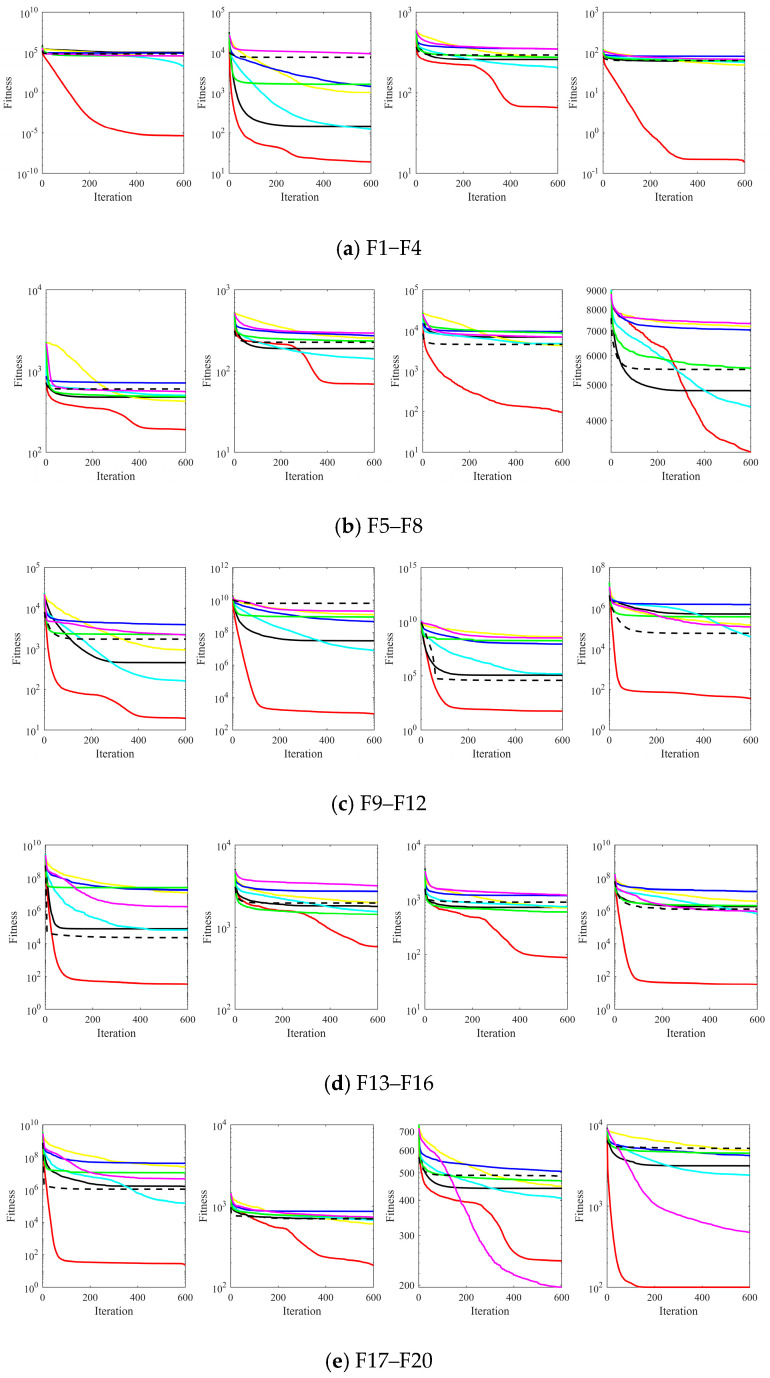
The mean value curves on CEC2017 functions.

**Figure 4 biomimetics-08-00348-f004:**
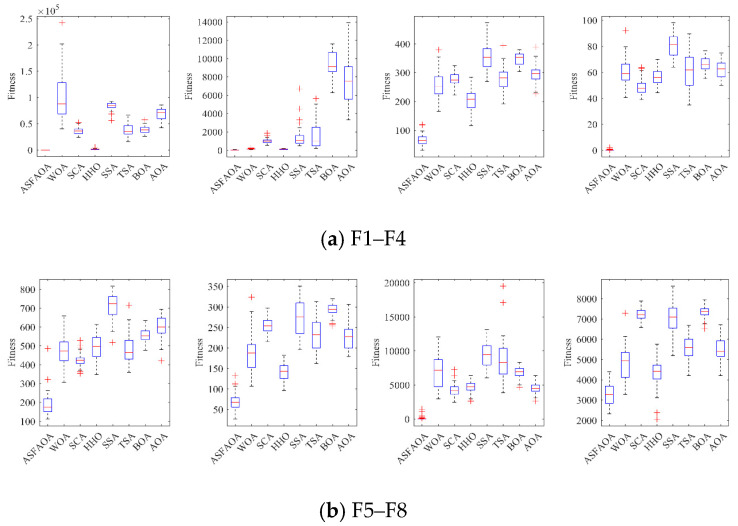
Box plot analysis for CEC2017 functions.

**Figure 5 biomimetics-08-00348-f005:**
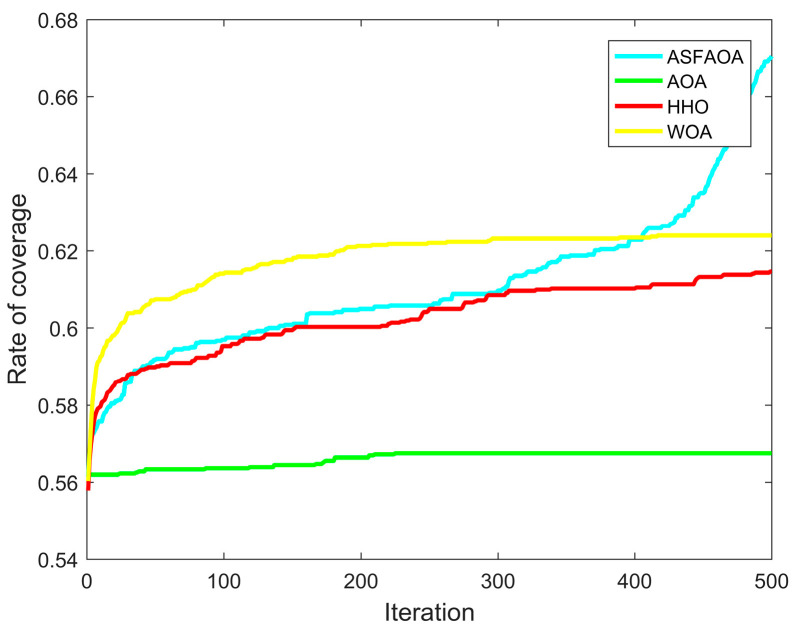
Coverage curves for case 1.

**Figure 6 biomimetics-08-00348-f006:**
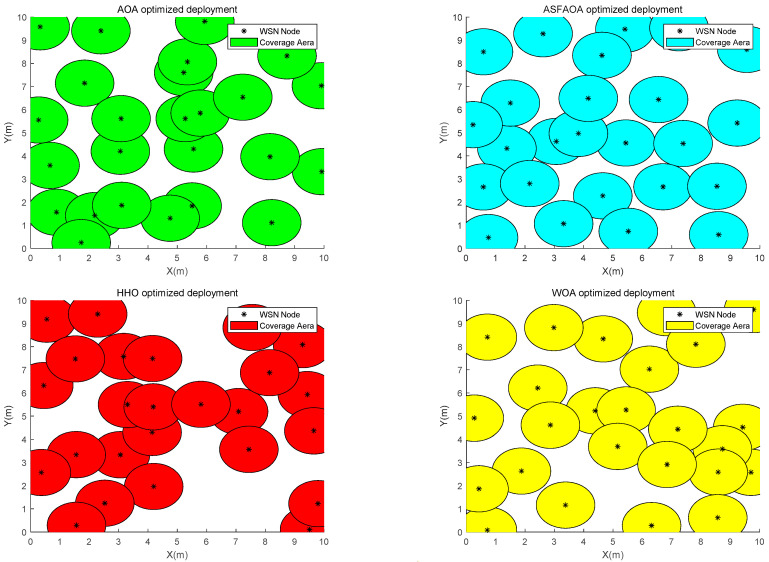
Node deployment in case 1.

**Figure 7 biomimetics-08-00348-f007:**
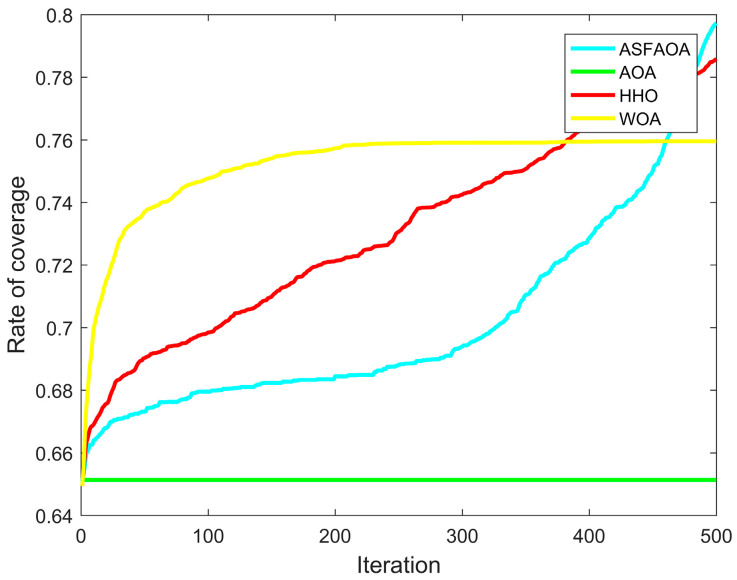
Coverage curves for case 2.

**Figure 8 biomimetics-08-00348-f008:**
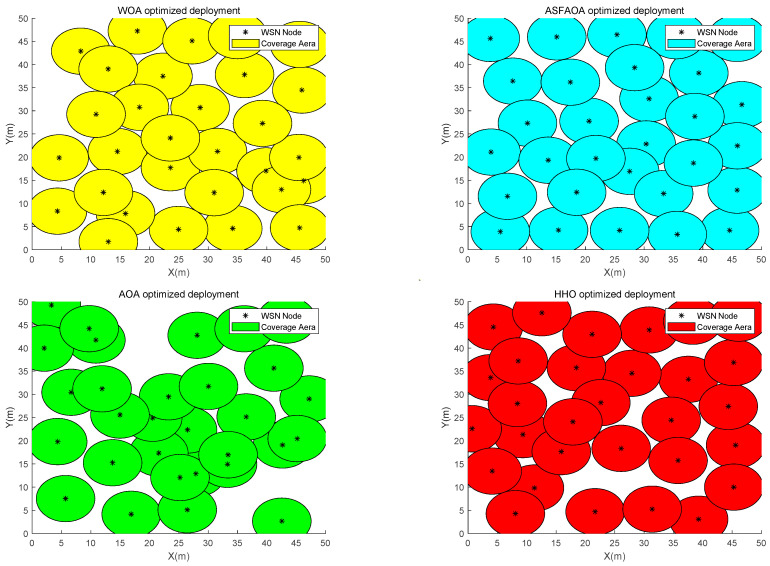
Node deployment in case 2.

**Table 1 biomimetics-08-00348-t001:** Descriptions of CEC 2017 test suite.

Type	No.	Description	Fi *
Unimodal functions	1	Shifted and Rotated Bent Cigar Function	300
Unimodal functions	2	Shifted and Rotated Rosenbrock’s Function	400
	3	Shifted and Rotated Rastrigin’s Function	500
	4	Shifted and Rotated Expanded Scaffer’s F6 Function	600
	5	Shifted and Rotated Lunacek Bi-Rastrigin Function	700
	6	Shifted and Rotated Non-Continuous Rastrigin’s Function	800
	7	Shifted and Rotated Levy Function	900
	8	Shifted and Rotated Schwefel’s Function	1000
Hybrid functions	9	Hybrid Function 1 (n = 3)	1100
	10	Hybrid Function 2 (n = 3)	1200
	11	Hybrid Function 3 (n = 3)	1300
	12	Hybrid Function 4 (n = 4)	1400
	13	Hybrid Function 5 (n = 4)	1500
	14	Hybrid Function 6 (n = 4)	1600
	15	Hybrid Function 6 (n = 5)	1700
	16	Hybrid Function 6 (n = 5)	1800
	17	Hybrid Function 6 (n = 5)	1900
	18	Hybrid Function 6 (n = 6)	2000
Composite functions	19	Composition Function 1 (n = 3)	2100
	20	Composition Function 2 (n = 3)	2200
	21	Composition Function 3 (n = 4)	2300
	22	Composition Function 4 (n = 4)	2400
	23	Composition Function 5 (n = 5)	2500
	24	Composition Function 6 (n = 5)	2600
	25	Composition Function 7 (n = 6)	2700
	26	Composition Function 8 (n = 6)	2800
	27	Composition Function 9 (n = 3)	2900
	28	Composition Function 10 (n = 3)	3000

**Table 2 biomimetics-08-00348-t002:** Parameter settings of compared algorithms.

Methods	Parameters
WOA	b=1,a=2 (Linearly decreased over iterations)
SCA	a=2 (Linearly decreased over iterations)
HHO	$\beta =1.5,E_{0}\in [-1,1]$
SSA	P=0.2,C=0.2
TSA	xr∈(1,4)
BOA	p=0.6,a=0.1,c=0.01
AOA	$Mop_{max}=1,Mop_{min}=0.2,C=1,\alpha =5,Mu=0.499$

**Table 3 biomimetics-08-00348-t003:** Results obtained with the methods for the CEC2017 test at D = 30.

Function	Items	ASFAOA	WOA	SCA	HHO	SSA	TSA	BOA	AOA
F1	Mean	4.51 × 10^−6^	1.02 × 10^5^	3.61 × 10^4^	1.68 × 10^3^	8.40 × 10^4^	3.83 × 10^4^	3.82 × 10^4^	6.91 × 10^4^
	Std	1.46 × 10^−6^	4.67 × 10^4^	6.47 × 10^3^	7.95 × 10^2^	6.59 × 10^3^	1.19 × 10^4^	6.97 × 10^3^	1.15 × 10^4^
	Rank	1	8	3	2	7	5	4	6
F2	Mean	1.93 × 10	1.46 × 10^2^	1.02 × 10^3^	1.23 × 10^2^	1.44 × 10^3^	1.62 × 10^3^	9.33 × 10^3^	7.61 × 10^3^
	Std	2.75 × 10	3.64 × 10	2.61 × 10^2^	3.33 × 10	1.09 × 10^3^	1.40 × 10^3^	1.29 × 10^3^	2.45 × 10^3^
	Rank	1	3	4	2	5	6	8	7
F3	Mean	6.55 × 10	2.57 × 10^2^	2.76 × 10^2^	2.05 × 10^2^	3.50 × 10^2^	2.76 × 10^2^	3.49 × 10^2^	2.95 × 10^2^
	Std	1.88 × 10	4.88 × 10	2.28 × 10	3.62 × 10	4.40 × 10	4.09 × 10	2.16 × 10	3.20 × 10
	Rank	1	3	4	2	8	5	7	6
F4	Mean	1.96 × 10^−1^	6.03 × 10	4.84 × 10	5.62 × 10	8.06 × 10	6.15 × 10	6.63 × 10	6.21 × 10
	Std	4.13 × 10^−1^	9.43 × 10^0^	5.55 × 10^0^	5.92 × 10^0^	8.84 × 10^0^	1.43 × 10	5.76 × 10^0^	6.71 × 10^0^
	Rank	1	4	2	3	8	5	7	6
F5	Mean	1.90 × 10^2^	4.76 × 10^2^	4.24 × 10^2^	4.98 × 10^2^	7.12 × 10^2^	4.83 × 10^2^	5.57 × 10^2^	6.00 × 10^2^
	Std	6.05 × 10	7.72 × 10	3.36 × 10	6.57 × 10	6.85 × 10	7.78 × 10	3.17 × 10	5.66 × 10
	Rank	1	3	2	5	8	4	6	7
F6	Mean	6.82 × 10	1.88 × 10^2^	2.54 × 10^2^	1.40 × 10^2^	2.72 × 10^2^	2.34 × 10^2^	2.93 × 10^2^	2.25 × 10^2^
	Std	2.07 × 10	4.52 × 10	1.89 × 10	2.13 × 10	4.31 × 10	3.99 × 10	1.54 × 10	2.67 × 10
	Rank	1	3	6	2	7	5	8	4
F7	Mean	9.71 × 10	6.83 × 10^3^	4.22 × 10^3^	4.69 × 10^3^	9.35 × 10^3^	8.57 × 10^3^	6.82 × 10^3^	4.50 × 10^3^
	Std	2.63 × 10^2^	2.35 × 10^3^	9.97 × 10^2^	8.28 × 10^2^	1.85 × 10^3^	3.01 × 10^3^	8.69 × 10^2^	7.24 × 10^2^
	Rank	1	6	2	4	8	7	5	3
F8	Mean	3.29 × 10^3^	4.82 × 10^3^	7.20 × 10^3^	4.35 × 10^3^	7.05 × 10^3^	5.55 × 10^3^	7.33 × 10^3^	5.51 × 10^3^
	Std	5.39 × 10^2^	8.20 × 10^2^	3.00 × 10^2^	7.25 × 10^2^	7.45 × 10^2^	6.07 × 10^2^	2.85 × 10^2^	5.83 × 10^2^
	Rank	1	3	7	2	6	5	8	4
F9	Mean	1.93 × 10	4.55 × 10^2^	9.42 × 10^2^	1.61 × 10^2^	3.91 × 10^3^	2.23 × 10^3^	2.19 × 10^3^	1.72 × 10^3^
	Std	1.70 × 10	1.40 × 10^2^	2.14 × 10^2^	4.86 × 10	1.64 × 10^3^	1.69 × 10^3^	6.72 × 10^2^	9.74 × 10^2^
	Rank	1	3	4	2	8	7	6	5
F10	Mean	1.00 × 10^3^	3.05 × 10^7^	1.26 × 10^9^	7.61 × 10^6^	4.69 × 10^8^	8.88 × 10^8^	2.08 × 10^9^	6.27 × 10^9^
	Std	2.76 × 10^2^	2.19 × 10^7^	3.15 × 10^8^	4.21 × 10^6^	3.76 × 10^8^	1.07 × 10^9^	7.43 × 10^8^	2.56 × 10^9^
	Rank	1	3	6	2	4	5	7	8
F11	Mean	5.51 × 10	1.14 × 10^5^	4.09 × 10^8^	1.51 × 10^5^	8.55 × 10^7^	1.75 × 10^8^	3.15 × 10^8^	3.80 × 10^4^
	Std	1.42 × 10	8.65 × 10^4^	1.50 × 10^8^	9.05 × 10^4^	4.66 × 10^8^	4.14 × 10^8^	2.10 × 10^8^	1.71 × 10^4^
	Rank	1	3	8	4	5	6	7	2
F12	Mean	3.52 × 10	5.12 × 10^5^	1.47 × 10^5^	3.82 × 10^4^	1.50 × 10^6^	3.73 × 10^5^	1.19 × 10^5^	5.72 × 10^4^
	Std	6.14 × 10^0^	5.25 × 10^5^	8.14 × 10^4^	4.25 × 10^4^	1.21 × 10^6^	6.73 × 10^5^	7.62 × 10^4^	4.92 × 10^4^
	Rank	1	7	5	2	8	6	4	3
F13	Mean	3.36 × 10	8.15 × 10^4^	1.29 × 10^7^	6.86 × 10^4^	1.83 × 10^7^	2.48 × 10^7^	1.82 × 10^6^	2.35 × 10^4^
	Std	1.18 × 10	3.82 × 10^4^	1.07 × 10^7^	4.86 × 10^4^	2.37 × 10^7^	7.80 × 10^7^	1.46 × 10^6^	1.22 × 10^4^
	Rank	1	4	6	3	7	8	5	2
F14	Mean	5.82 × 10^2^	1.79 × 10^3^	2.01 × 10^3^	1.55 × 10^3^	2.74 × 10^3^	1.43 × 10^3^	3.18 × 10^3^	1.98 × 10^3^
	Std	2.51 × 10^2^	4.36 × 10^2^	2.98 × 10^2^	3.56 × 10^2^	5.38 × 10^2^	2.92 × 10^2^	4.12 × 10^2^	5.09 × 10^2^
	Rank	1	4	6	3	7	2	8	5
F15	Mean	8.75 × 10	7.32 × 10^2^	7.16 × 10^2^	7.48 × 10^2^	1.20 × 10^3^	6.06 × 10^2^	1.22 × 10^3^	9.12 × 10^2^
	Std	4.91 × 10	2.68 × 10^2^	1.75 × 10^2^	2.19 × 10^2^	3.85 × 10^2^	2.30 × 10^2^	2.49 × 10^2^	2.67 × 10^2^
	Rank	1	4	3	5	7	2	8	6
F16	Mean	3.27 × 10	1.84 × 10^6^	3.93 × 10^6^	6.90 × 10^5^	1.51 × 10^7^	2.08 × 10^6^	9.60 × 10^5^	1.29 × 10^6^
	Std	2.83 × 10^0^	2.09 × 10^6^	3.32 × 10^6^	8.77 × 10^5^	1.51 × 10^7^	4.09 × 10^6^	6.22 × 10^5^	1.60 × 10^6^
	Rank	1	5	7	2	8	6	3	4
F17	Mean	2.39 × 10	1.60 × 10^6^	2.56 × 10^7^	1.46 × 10^5^	4.23 × 10^7^	1.11 × 10^7^	4.61 × 10^6^	1.08 × 10^6^
	Std	3.26 × 10^0^	1.36 × 10^6^	1.31 × 10^7^	1.42 × 10^5^	1.23 × 10^8^	3.45 × 10^7^	4.06 × 10^6^	1.39 × 10^5^
	Rank	1	4	7	2	8	6	5	3
F18	Mean	1.87 × 10^2^	7.03 × 10^2^	6.05 × 10^2^	6.71 × 10^2^	8.59 × 10^2^	7.24 × 10^2^	7.29 × 10^2^	6.94 × 10^2^
	Std	8.81 × 10	1.96 × 10^2^	1.32 × 10^2^	2.01 × 10^2^	2.42 × 10^2^	2.09 × 10^2^	9.88 × 10	1.54 × 10^2^
	Rank	1	5	2	3	8	6	7	4
F19	Mean	2.44 × 10^2^	4.40 × 10^2^	4.48 × 10^2^	4.06 × 10^2^	5.06 × 10^2^	4.68 × 10^2^	1.97 × 10^2^	4.87 × 10^2^
	Std	1.35 × 10	4.86 × 10	1.97 × 10	3.51 × 10	5.36 × 10	4.96 × 10	3.01 × 10	5.23 × 10
	Rank	2	4	5	3	8	6	1	7
F20	Mean	1.00 × 10^2^	3.13 × 10^3^	4.85 × 10^3^	2.39 × 10^3^	4.18 × 10^3^	4.47 × 10^3^	4.71 × 10^2^	5.13 × 10^3^
	Std	7.09 × 10^−6^	2.44 × 10^3^	2.94 × 10^3^	2.37 × 10^3^	1.88 × 10^3^	2.09 × 10^3^	7.76 × 10	1.21 × 10^3^
	Rank	1	4	7	3	5	6	2	8
F21	Mean	3.86 × 10^2^	7.09 × 10^2^	6.84 × 10^2^	7.05 × 10^2^	8.60 × 10^2^	7.86 × 10^2^	6.97 × 10^2^	9.68 × 10^2^
	Std	1.57 × 10	8.66 × 10	3.49 × 10	7.35 × 10	1.00 × 10^2^	8.15 × 10	5.59 × 10	9.10 × 10
	Rank	1	5	2	4	7	6	3	8
F22	Mean	4.41 × 10^2^	7.30 × 10^2^	7.51 × 10^2^	8.26 × 10^2^	8.99 × 10^2^	8.47 × 10^2^	1.10 × 10^3^	1.14 × 10^3^
	Std	2.72 × 10	7.30 × 10	2.52 × 10	7.42 × 10	1.37 × 10^2^	8.08 × 10	1.68 × 10^2^	1.09 × 10^2^
	Rank	1	2	3	4	6	5	7	8
F23	Mean	3.88 × 10^2^	4.66 × 10^2^	6.99 × 10^2^	4.11 × 10^2^	7.84 × 10^2^	7.61 × 10^2^	1.75 × 10^3^	1.67 × 10^3^
	Std	3.75 × 10^0^	3.26 × 10	5.73 × 10	1.87 × 10	1.30 × 10^2^	3.02 × 10^2^	2.01 × 10^2^	4.55 × 10^2^
	Rank	1	3	4	2	6	5	8	7
F24	Mean	1.20 × 10^3^	4.44 × 10^3^	4.24 × 10^3^	3.94 × 10^3^	6.30 × 10^3^	5.01 × 10^3^	5.21 × 10^3^	6.40 × 10^3^
	Std	5.80 × 10^2^	1.11 × 10^3^	2.93 × 10^2^	1.10 × 10^3^	1.11 × 10^3^	8.76 × 10^2^	1.49 × 10^3^	7.22 × 10^2^
	Rank	1	4	3	2	7	5	6	8
F25	Mean	4.84 × 10^2^	6.47 × 10^2^	7.03 × 10^2^	6.05 × 10^2^	9.56 × 10^2^	7.30 × 10^2^	8.14 × 10^2^	1.34 × 10^3^
	Std	1.30 × 10	8.42 × 10	3.63 × 10	4.00 × 10	1.65 × 10^2^	9.92 × 10	9.81 × 10	2.14 × 10^2^
	Rank	1	3	4	2	7	5	6	8
F26	Mean	3.30 × 10^2^	5.13 × 10^2^	1.04 × 10^3^	4.62 × 10^2^	1.06 × 10^3^	1.27 × 10^3^	3.28 × 10^3^	2.95 × 10^3^
	Std	5.09 × 10	3.27 × 10	1.23 × 10^2^	2.60 × 10	3.22 × 10^2^	4.52 × 10^2^	3.99 × 10^2^	6.15 × 10^2^
	Rank	1	3	4	2	5	6	8	7
F27	Mean	5.82 × 10^2^	1.88 × 10^3^	1.70 × 10^3^	1.32 × 10^3^	2.64 × 10^3^	1.58 × 10^3^	3.04 × 10^3^	2.43 × 10^3^
	Std	6.25 × 10	4.08 × 10^2^	2.31 × 10^2^	2.56 × 10^2^	6.35 × 10^2^	4.08 × 10^2^	4.72 × 10^2^	5.22 × 10^2^
	Rank	1	5	4	2	7	3	8	6
F28	Mean	2.01 × 10^3^	7.04 × 10^6^	7.41 × 10^7^	1.01 × 10^6^	4.85 × 10^7^	1.33 × 10^7^	3.98 × 10^7^	1.47 × 10^7^
	Std	3.41 × 10	4.69 × 10^6^	3.63 × 10^7^	6.08 × 10^5^	3.75 × 10^7^	1.07 × 10^7^	2.31 × 10^7^	1.01 × 10^7^
	Rank	1	3	8	2	7	4	6	5
Average ranking		1.04	3.96	4.57	2.71	6.86	5.25	6.00	5.61
Total ranking		1	3	4	2	8	5	7	6

**Table 4 biomimetics-08-00348-t004:** The *p*-value results on the CEC 2017 30D test obtained with the Wilcoxon signed-rank test.

ASFAOA vs.	WOA		SCA		HHO		SSA		TSA		BOA		AOA	
No.	*p*-Value	Win	*p*-Value	Win	*p*-Value	Win	*p*-Value	Win	*p*-Value	Win	*p*-Value	Win	*p*-Value	Win
F1	5.15 × 10^−10^	−	5.15 × 10^−10^	−	5.15 × 10^−10^	−	5.15 × 10^−10^	−	5.15 × 10^−10^	−	5.15 × 10^−10^	−	5.15 × 10^−10^	−
F2	5.15 × 10^−10^	−	5.15 × 10^−10^	−	5.15 × 10^−10^	−	5.15 × 10^−10^	−	5.15 × 10^−10^	−	5.15 × 10^−10^	−	5.15 × 10^−10^	−
F3	5.15 × 10^−10^	−	5.15 × 10^−10^	−	5.15 × 10^−10^	−	5.15 × 10^−10^	−	5.15 × 10^−10^	−	5.15 × 10^−10^	−	5.15 × 10^−10^	−
F4	5.15 × 10^−10^	−	5.15 × 10^−10^	−	5.15 × 10^−10^	−	5.15 × 10^−10^	−	5.15 × 10^−10^	−	5.15 × 10^−10^	−	5.15 × 10^−10^	−
F5	5.15 × 10^−10^	−	5.46 × 10^−10^	−	5.15 × 10^−10^	−	5.15 × 10^−10^	−	5.46 × 10^−10^	−	5.15 × 10^−10^	−	5.15 × 10^−10^	−
F6	5.46 × 10^−10^	−	5.15 × 10^−10^	−	5.15 × 10^−10^	−	5.15 × 10^−10^	−	5.15 × 10^−10^	−	5.15 × 10^−10^	−	5.15 × 10^−10^	−
F7	5.15 × 10^−10^	−	5.15 × 10^−10^	−	5.15 × 10^−10^	−	5.15 × 10^−10^	−	5.15 × 10^−10^	−	5.15 × 10^−10^	−	5.15 × 10^−10^	−
F8	1.32 × 10^−9^	−	5.15 × 10^−10^	−	4.17 × 10^−8^	−	5.15 × 10^−10^	−	5.15 × 10^−10^	−	5.15 × 10^−10^	−	5.15 × 10^−10^	−
F9	5.15 × 10^−10^	−	5.15 × 10^−10^	−	5.15 × 10^−10^	−	5.15 × 10^−10^	−	5.15 × 10^−10^	−	5.15 × 10^−10^	−	5.15 × 10^−10^	−
F10	5.15 × 10^−10^	−	5.15 × 10^−10^	−	5.15 × 10^−10^	−	5.15 × 10^−10^	−	5.15 × 10^−10^	−	5.15 × 10^−10^	−	5.15 × 10^−10^	−
F11	5.15 × 10^−10^	−	5.15 × 10^−10^	−	5.15 × 10^−10^	−	5.15 × 10^−10^	−	5.15 × 10^−10^	−	5.15 × 10^−10^	−	5.15 × 10^−10^	−
F12	5.15 × 10^−10^	−	5.15 × 10^−10^	−	5.15 × 10^−10^	−	5.15 × 10^−10^	−	5.15 × 10^−10^	−	5.15 × 10^−10^	−	5.15 × 10^−10^	−
F13	5.15 × 10^−10^	−	5.15 × 10^−10^	−	5.15 × 10^−10^	−	5.15 × 10^−10^	−	5.15 × 10^−10^	−	5.15 × 10^−10^	−	5.15 × 10^−10^	−
F14	5.15 × 10^−10^	−	5.15 × 10^−10^	−	5.15 × 10^−10^	−	5.15 × 10^−10^	−	9.87 × 10^−10^	−	5.15 × 10^−10^	−	5.15 × 10^−10^	−
F15	5.15 × 10^−10^	−	5.15 × 10^−10^	−	5.15 × 10^−10^	−	5.15 × 10^−10^	−	5.15 × 10^−10^	−	5.15 × 10^−10^	−	5.15 × 10^−10^	−
F16	5.15 × 10^−10^	−	5.15 × 10^−10^	−	5.15 × 10^−10^	−	5.15 × 10^−10^	−	5.15 × 10^−10^	−	5.15 × 10^−10^	−	5.15 × 10^−10^	−
F17	5.15 × 10^−10^	−	5.15 × 10^−10^	−	5.15 × 10^−10^	−	5.15 × 10^−10^	−	5.15 × 10^−10^	−	5.15 × 10^−10^	−	5.15 × 10^−10^	−
F18	5.15 × 10^−10^	−	5.15 × 10^−10^	−	5.15 × 10^−10^	−	5.15 × 10^−10^	−	5.15 × 10^−10^	−	5.15 × 10^−10^	−	5.46 × 10^−10^	−
F19	5.15 × 10^−10^	−	5.15 × 10^−10^	−	5.15 × 10^−10^	−	5.15 × 10^−10^	−	5.15 × 10^−10^	−	8.18 × 10^−9^	+	5.15 × 10^−10^	−
F20	5.15 × 10^−10^	−	5.15 × 10^−10^	−	5.15 × 10^−10^	−	5.15 × 10^−10^	−	5.15 × 10^−10^	−	5.15 × 10^−10^	−	5.15 × 10^−10^	−
F21	5.15 × 10^−10^	−	5.15 × 10^−10^	−	5.15 × 10^−10^	−	5.15 × 10^−10^	−	5.15 × 10^−10^	−	5.15 × 10^−10^	−	5.15 × 10^−10^	−
F22	5.15 × 10^−10^	−	5.15 × 10^−10^	−	5.15 × 10^−10^	−	5.15 × 10^−10^	−	5.15 × 10^−10^	−	5.15 × 10^−10^	−	5.15 × 10^−10^	−
F23	5.15 × 10^−10^	−	5.15 × 10^−10^	−	1.86 × 10^−10^	−	5.15 × 10^−10^	−	5.15 × 10^−10^	−	5.15 × 10^−10^	−	5.15 × 10^−10^	−
F24	7.35 × 10^−10^	−	5.15 × 10^−10^	−	8.27 × 10^−10^	−	5.15 × 10^−10^	−	5.15 × 10^−10^	−	5.15 × 10^−10^	−	5.15 × 10^−10^	−
F25	5.15 × 10^−10^	−	5.15 × 10^−10^	−	5.15 × 10^−10^	−	5.15 × 10^−10^	−	5.15 × 10^−10^	−	5.15 × 10^−10^	−	5.15 × 10^−10^	−
F26	5.15 × 10^−10^	−	5.15 × 10^−10^	−	5.15 × 10^−10^	−	5.15 × 10^−10^	−	5.15 × 10^−10^	−	5.15 × 10^−10^	−	5.15 × 10^−10^	−
F27	5.15 × 10^−10^	−	5.15 × 10^−10^	−	5.15 × 10^−10^	−	5.15 × 10^−10^	−	5.15 × 10^−10^	−	5.15 × 10^−10^	−	5.15 × 10^−10^	−
F28	5.15 × 10^−10^	−	5.15 × 10^−10^	−	5.15 × 10^−10^	−	5.15 × 10^−10^	−	5.15 × 10^−10^	−	5.15 × 10^−10^	−	5.15 × 10^−10^	−
+/−/=	28/0/0		28/0/0		28/0/0		28/0/0		28/0/0		27/0/1		28/0/0	

**Table 5 biomimetics-08-00348-t005:** ASFAOA variants with different improvement strategies.

Algorithm	DOL	ASS	ACA	ODE
ASFAOA-1	Yes	No	No	No
ASFAOA-2	No	Yes	No	No
ASFAOA-3	No	No	Yes	No
ASFAOA-4	No	No	No	Yes
ASFAOA	Yes	Yes	Yes	Yes

**Table 6 biomimetics-08-00348-t006:** Statistics of the results in CEC2017 30D test using ASFAOA variants.

Function	Items	ASFAOA	ASFAOA-1	ASFAOA-2	ASFAOA-3	ASFAOA-4	AOA
F1	Mean	4.51 × 10^−6^	7.36 × 10^4^	5.76 × 10^4^	1.13 × 10^−5^	8.03 × 10^4^	6.91 × 10^4^
	Std	1.46 × 10^−6^	9.22 × 10^3^	9.54 × 10^3^	1.35 × 10^−6^	9.21 × 10^3^	1.15 × 10^4^
	Rank	1	5	3	2	6	4
F2	Mean	1.93 × 10	1.41 × 10^3^	5.77 × 10^2^	2.66 × 10	1.79 × 10^3^	7.61 × 10^3^
	Std	2.75 × 10	9.71 × 10^2^	3.31 × 10^2^	3.55 × 10	1.11 × 10^3^	2.45 × 10^3^
	Rank	1	4	3	2	5	6
F3	Mean	6.55 × 10	2.70 × 10^2^	2.49 × 10^2^	2.40 × 10^2^	2.75 × 10^2^	2.95 × 10^2^
	Std	1.88 × 10	3.92 × 10	4.32 × 10	4.34 × 10	3.35 × 10	3.20 × 10
	Rank	1	4	3	2	5	6
F4	Mean	1.96 × 10^−1^	3.59 × 10	5.76 × 10	5.21 × 10	3.43 × 10	6.21 × 10
	Std	4.13 × 10^−1^	4.71 × 10^0^	9.30 × 10^0^	7.33 × 10^0^	4.74 × 10^0^	6.71 × 10^0^
	Rank	1	3	5	4	2	6
F5	Mean	1.90 × 10^2^	4.38 × 10^2^	5.89 × 10^2^	5.78 × 10^2^	4.32 × 10^2^	6.00 × 10^2^
	Std	6.05 × 10	5.90 × 10	6.33 × 10	4.90 × 10	6.87 × 10	5.66 × 10
	Rank	1	3	5	4	2	6
F6	Mean	6.82 × 10	2.18 × 10^2^	1.84 × 10^2^	1.69 × 10^2^	2.16 × 10^2^	2.25 × 10^2^
	Std	2.07 × 10	2.95 × 10	3.44 × 10	3.01 × 10	2.46 × 10	2.67 × 10
	Rank	1	5	3	2	4	6
F7	Mean	9.71 × 10	4.56 × 10^3^	4.62 × 10^3^	4.70 × 10^3^	4.43 × 10^3^	4.50 × 10^3^
	Std	2.63 × 10^2^	6.71 × 10^2^	5.02 × 10^2^	1.28 × 10^3^	5.56 × 10^2^	7.24 × 10^2^
	Rank	1	4	5	6	2	3
F8	Mean	3.29 × 10^3^	5.02 × 10^3^	3.81 × 10^3^	4.14 × 10^3^	4.93 × 10^3^	5.51 × 10^3^
	Std	5.39 × 10^2^	4.20 × 10^2^	5.78 × 10^2^	5.74 × 10^2^	4.53 × 10^2^	5.83 × 10^2^
	Rank	1	5	2	3	4	6
F9	Mean	1.93 × 10	2.37 × 10^3^	4.92 × 10^2^	8.94 × 10	2.65 × 10^3^	1.72 × 10^3^
	Std	1.70 × 10	1.00 × 10^3^	3.88 × 10^2^	2.69 × 10	1.44 × 10^3^	9.74 × 10^2^
	Rank	1	5	3	2	6	4
F10	Mean	1.00 × 10^3^	1.08 × 10^9^	3.22 × 10^7^	2.29 × 10^3^	1.11 × 10^9^	6.27 × 10^9^
	Std	2.76 × 10^2^	8.19 × 10^8^	3.98 × 10^7^	1.58 × 10^3^	8.13 × 10^8^	2.56 × 10^9^
	Rank	1	4	3	2	5	6
F11	Mean	5.51 × 10	2.33 × 10^7^	1.99 × 10^4^	1.18 × 10^3^	2.95 × 10^7^	3.80 × 10^4^
	Std	1.42 × 10	3.57 × 10^7^	1.33 × 10^4^	6.98 × 10^2^	4.15 × 10^7^	1.71 × 10^4^
	Rank	1	5	3	2	6	4
F12	Mean	3.52 × 10	5.32 × 10^5^	7.15 × 10^4^	8.18 × 10	4.71 × 10^5^	5.72 × 10^4^
	Std	6.14 × 10^0^	5.15 × 10^5^	6.52 × 10^4^	1.81 × 10	4.02 × 10^5^	4.92 × 10^4^
	Rank	1	6	4	2	5	3
F13	Mean	3.36 × 10	5.78 × 10^3^	8.96 × 10^3^	2.35 × 10^2^	6.13 × 10^3^	2.35 × 10^4^
	Std	1.18 × 10	5.79 × 10^3^	6.46 × 10^3^	9.44 × 10	6.06 × 10^3^	1.22 × 10^4^
	Rank	1	3	5	2	4	6
F14	Mean	5.82 × 10^2^	1.90 × 10^3^	1.44 × 10^3^	1.28 × 10^3^	1.83 × 10^3^	1.98 × 10^3^
	Std	2.51 × 10^2^	3.36 × 10^2^	3.66 × 10^2^	2.92 × 10^2^	3.83 × 10^2^	5.09 × 10^2^
	Rank	1	5	3	2	4	6
F15	Mean	8.75 × 10	6.81 × 10^2^	5.81 × 10^2^	6.48 × 10^2^	6.52 × 10^2^	9.12 × 10^2^
	Std	4.91 × 10	2.10 × 10^2^	2.07 × 10^2^	2.46 × 10^2^	2.23 × 10^2^	2.67 × 10^2^
	Rank	1	5	2	3	4	6
F16	Mean	3.27 × 10	9.02 × 10^5^	6.42 × 10^5^	6.03 × 10	9.57 × 10^5^	1.29 × 10^6^
	Std	2.83 × 10^0^	5.80 × 10^5^	1.32 × 10^6^	6.55 × 10	7.72 × 10^5^	1.60 × 10^6^
	Rank	1	4	3	2	5	6
F17	Mean	2.39 × 10	5.26 × 10^3^	9.59 × 10^3^	5.60 × 10	5.28 × 10^3^	1.08 × 10^6^
	Std	3.26 × 10^0^	1.02 × 10^4^	1.05 × 10^4^	2.36 × 10	9.15 × 10^3^	1.39 × 10^5^
	Rank	1	3	5	2	4	6
F18	Mean	1.87 × 10^2^	5.92 × 10^2^	5.27 × 10^2^	5.78 × 10^2^	5.69 × 10^2^	6.94 × 10^2^
	Std	8.81 × 10	1.74 × 10^2^	1.74 × 10^2^	1.60 × 10^2^	1.79 × 10^2^	1.54 × 10^2^
	Rank	1	5	2	4	3	6
F19	Mean	2.44 × 10^2^	3.78 × 10^2^	3.97 × 10^2^	4.03 × 10^2^	3.98 × 10^2^	4.87 × 10^2^
	Std	1.35 × 10	1.11 × 10^2^	4.06 × 10	4.69 × 10	1.00 × 10^2^	5.23 × 10
	Rank	1	2	3	5	4	6
F20	Mean	1.00 × 10^2^	3.18 × 10^3^	1.27 × 10^3^	3.29 × 10^3^	4.06 × 10^3^	5.13 × 10^3^
	Std	7.09 × 10^−6^	1.81 × 10^3^	1.04 × 10^3^	2.06 × 10^3^	1.81 × 10^3^	1.21 × 10^3^
	Rank	1	3	2	4	5	6
F21	Mean	3.86 × 10^2^	6.91 × 10^2^	6.65 × 10^2^	8.05 × 10^2^	6.73 × 10^2^	9.68 × 10^2^
	Std	1.57 × 10	6.34 × 10	6.69 × 10	7.90 × 10	5.88 × 10	9.10 × 10
	Rank	1	4	2	5	3	6
F22	Mean	4.41 × 10^2^	9.00 × 10^2^	7.37 × 10^2^	9.72 × 10^2^	9.21 × 10^2^	1.14 × 10^3^
	Std	2.72 × 10	7.59 × 10	6.56 × 10	9.31 × 10	7.68 × 10	1.09 × 10^2^
	Rank	1	3	2	5	4	6
F23	Mean	3.88 × 10^2^	7.62 × 10^2^	5.98 × 10^2^	4.35 × 10^2^	7.83 × 10^2^	1.67 × 10^3^
	Std	3.75 × 10^0^	1.47 × 10^2^	7.47 × 10	2.28 × 10	2.12 × 10^2^	4.55 × 10^2^
	Rank	1	4	3	2	5	6
F24	Mean	1.20 × 10^3^	4.18 × 10^3^	4.15 × 10^3^	4.78 × 10^3^	3.88 × 10^3^	6.40 × 10^3^
	Std	5.80 × 10^2^	1.25 × 10^3^	1.25 × 10^3^	2.29 × 10^3^	1.30 × 10^3^	7.22 × 10^2^
	Rank	1	4	3	5	2	6
F25	Mean	4.84 × 10^2^	7.02 × 10^2^	7.85 × 10^2^	9.52 × 10^2^	7.19 × 10^2^	1.34 × 10^3^
	Std	1.30 × 10	7.30 × 10	9.63 × 10	1.52 × 10^2^	8.94 × 10	2.14 × 10^2^
	Rank	1	2	4	5	3	6
F26	Mean	3.30 × 10^2^	1.13 × 10^3^	7.79 × 10^2^	3.09 × 10^2^	1.16 × 10^3^	2.95 × 10^3^
	Std	5.09 × 10	2.65 × 10^2^	1.95 × 10^2^	3.14 × 10	3.17 × 10^2^	6.15 × 10^2^
	Rank	2	4	3	1	5	6
F27	Mean	5.82 × 10^2^	1.56 × 10^3^	1.59 × 10^3^	1.44 × 10^3^	1.55 × 10^3^	2.43 × 10^3^
	Std	6.25 × 10	3.45 × 10^2^	3.01 × 10^2^	2.78 × 10^2^	3.66 × 10^2^	5.22 × 10^2^
	Rank	1	4	5	2	3	6
F28	Mean	2.01 × 10^3^	6.24 × 10^6^	5.05 × 10^5^	2.84 × 10^3^	5.72 × 10^6^	1.47 × 10^7^
	Std	3.41 × 10	5.88 × 10^6^	1.30 × 10^6^	4.69 × 10^2^	6.03 × 10^6^	1.01 × 10^7^
	Rank	1	5	3	2	4	6
Average ranking		1.04	4.04	3.29	3.00	4.07	5.57
Total ranking		1	4	3	2	5	6

**Table 7 biomimetics-08-00348-t007:** Comparison of WSN performance in case 1.

Algorithm	Best	Mean	Std
ASFAOA	75.21%	67.05%	0.03
AOA	59.50%	56.75%	0.01
HHO	66.94%	61.46%	0.02
WOA	66.94%	62.40%	0.03

**Table 8 biomimetics-08-00348-t008:** Comparison of WSN performance in case 2.

Algorithm	Best	Mean	Std
ASFAOA	83.93%	79.72%	0.03
AOA	69.40%	65.14%	0.02
HHO	78.82%	78.58%	0.01
WOA	78.82%	75.95%	0.02

## Data Availability

The data presented in this study are available on request from the corresponding author.

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
