# Peer review of "A Multiple Mechanism Enhanced Arithmetic Optimization Algorithm for Numerical Problems"

_biomimetics, 2023, doi:10.3390/biomimetics8040348_

Round 1
Reviewer 1 Report
I have the following comments:
- The abstract does not reflect the performed work completely. Please update it so that it can comprehensively cover what you have done and what specific results you obtained.
- Not all reported variants of the AOA have been covered. The authors must have a look at the following variants and discuss them within the manuscript. Then they need to explain the difference of their proposal.
· https://doi.org/10.1080/02286203.2023.2196736
· https://doi.org/10.1007/s00500-022-07068-x
· https://doi.org/10.1007/s12530-021-09402-4
· https://doi.org/10.1007/978-981-19-0332-8_20
· https://dx.doi.org/10.14744/sigma.2022.00056
· https://doi.org/10.1109/ISMSIT52890.2021.9604531
- The improvement method in the following studies have similarities to some extent. It is better to specifically explain the difference of this study so that the readers can appreciate the contribution of this work.
· Logarithmic spiral search based arithmetic optimization algorithm with selective mechanism and its application to functional electrical stimulation system control
· Augmented hunger games search algorithm using logarithmic spiral opposition-based learning for function optimization and controller design
- For the simulation of WSN, more additional recent algorithms may be preferred for comparisons.
- The conclusion should be rewritten to better summarise the performed work and the future scope.
- The reason of choosing CEC2017 benchmark functions must clearly be stated, otherwise one of the more challenging CEC functions should be preferred.
Apart from the above listed concerns, the study is good in general. Good luck with the revision.
Minor grammar related issues and typos should be corrected.
Reviewer 2 Report
Please see the attached file as Reviewer's comments.

Minor grammatical mistakes are there at some places.
Reviewer 3 Report
1. There are some writing irregularities in the manuscript, such as Figure [1]. Please check the manuscript to revise them.
2. In the manuscript, authors propose four strategies to enhance the performance of the AOA algorithm, but how do these four strategies enhance the performance of the algorithm? Suggestion:
1) Supplement experiments for each strategy to verify that each strategy employed indeed enhances the performance of the AOA algorithm;
2) Provide a detailed discussion on how to enhance the performance of the AOA algorithm.
3. The manuscript uses four strategies to enhance the performance of the AOA algorithm, which may greatly increase the time complexity of the proposed algorithm. Therefore, it is necessary to analyze the time complexity of the proposed algorithm.
4. How does the Adaptive cosine acceleration function balance the global exploration and local exploitation capabilities of the proposed algorithm? Please provide a detailed explanation.
5. There are many parameters involved in the proposed algorithm, such as parameter k in the Double position learning strategy. These parameters have an impact on the performance improvement of the proposed algorithm. Therefore, sensitivity analysis of important parameters in the algorithm is also necessary.
6.The conclusion section is too simple to summarize the work of this manuscript, and it is recommended to rewrite it.
English writing needs further polished.
Reviewer 4 Report
Authors propose an Arithmetic Optimization Algorithm variant called ASFAOA. It integrates double-opposite learning mechanism, an adaptive spiral search strategy, an offset distribution estimation strategy, and a modified cosine acceleration function formula into the original AOA.
It is an interesting paper although there are some issues that must be fixed:
In page, 6, authors name Figure [1]. It should be Figure 1.
In Figure 1, authors must add the title of the X and Y axes and add their units.
Authors must explain better each graph included in Figure 3 and Figure 4. One by one.
I have found a related work on coverage optimization that should be cited:
CASMOC: a novel complex alliance strategy with multi-objective optimization of coverage in wireless sensor networks, wireless Networks 23, 1201-1222. 2017
Authors should cite the simulator used for their experiments.
Round 2
Reviewer 3 Report
The manuscript can be accepted.
English language needs to be polished further.